# OpenReview forum: "COLD-Steer: Steering Large Language Models via In-Context One-step Learning Dynamics"
_ICLR.cc/2026/Conference — ICLR 2026 Poster_

### Official Review · Reviewer_QoRx · 2025-10-31

**Soundness:** 3
**Presentation:** 2
**Contribution:** 3
**Rating:** 6
**Confidence:** 3

**Summary:**

The authors proposed a training-free activation steering method, termed COLD-Steer. The core idea is that in-context examples of a desired behavior implicitly define the direction of gradient descent in activation space; thus, by simulating this “one-step learning dynamic,” the model can be “steered” without retraining.

**Strengths:**

1. The theory bridge activation steering with learning dynamics is elegant. It unifies contrastive (CAA, DiffMean) and parameter-tuning (ReFT) perspectives under one gradient-based formulation.

2. The authors performed broad evaluations which spans multiple LLMs and diverse downstream tasks (e.g., bias mitigation, hallucination reduction, refusal, sycophancy, pluralistic alignment). The COLD-Steer also shows competitive accuracy than the baselines.

3. Avoids expensive backpropagation during steering with 10–50× fewer labeled examples.

**Weaknesses:**

1. Limited theoretical rigor in approximations. The unit kernel assumption (κ = 1) oversimplifies eNTK behavior and may obscure causality.

2. Some important details are missing. No clear separation of effects from η (steering magnitude) or layer choice; lack of sensitivity or robustness testing.

3. Most results are on small/medium LLMs (7B). No evidence the method scales to larger scale-level or multi-modal models.

4. COLD-Steer relies on in-context examples to approximate the “one-step learning dynamics.” This inherently depends on the number and quality of examples that can fit into the context window (ICL window). Current LLMs (e.g., Llama-2-7B-chat) have a limited token context.

**Questions:**

Q1. Include layer-wise ablation is necessary, which layers yield maximal steerability vs. stability?

Q2. The authors acknowledge this limitation briefly (“future work should develop more sophisticated approximations of the neural tangent kernel”) but provide no empirical study on how κ or layer l affect the approximation quality.

---

> ### Author Response · Authors · 2025-11-21
> **Author Response to Reviewer QoRx (1/3)**
>
> We are delighted by the reviewer’s positive support of our work as they find our approach elegant and efficient, and our experiments comprehensive. They also raise very valuable concerns that help us improve parts of the exposition. We provide our responses to those concerns below and will also revise our paper to incorporate our discussions.
>
> > Limited theoretical rigor in approximations. The unit kernel assumption (κ = 1) oversimplifies eNTK behavior and may obscure causality.
>
>
> We thank the reviewer for giving us an opportunity to investigate this approximation further. In particular, we investigate why a unit kernel can be effective in general. Specifically, we observe that the approximation $\langle \nabla_{\theta} Z(x_i), \nabla_{\theta} Z(x_j) \rangle \approx 1$ can hold when the parameter gradients of a subnetwork are approximately the same (up to scaling) across inputs in a dataset. This occurs when the per-example gradient vectors are highly aligned or dominated by a single common direction. In such cases, each entry is roughly equal to the product of two similar norms, and after normalization, the resulting kernel closely resembles a unit (all-ones) kernel. Since all the inputs in the dataset are designed to elicit the same underlying behavior, we can expect the gradients with respect to the model’s parameters to be highly aligned. This is based on the assumption that the model internally encodes the relevant high-level concept (such as specific behaviors) in a consistent and linear way across inputs, which is often called the linear representation hypothesis [1,2]. In other words, the directions in activation or gradient space corresponding to a particular concept are similar across different inputs that express the concept. This explains why the kernel, computed as the inner product of per-example gradients, can be well-approximated by a unit (all-ones) matrix: each input contributes a gradient pointing along the same underlying conceptual direction, making them appear nearly identical in the kernel space after normalization.
>
>
>
>
> > Some important details are missing. No clear separation of effects from η (steering magnitude) or layer choice; lack of sensitivity or robustness testing.
> >
> > Include layer-wise ablation is necessary, which layers yield maximal steerability vs. stability?
>
>
> We analyze the sensitivity of COLD-Steer with respect to the choice of steering layer and $\eta$. In particular, we consider the Llama-2-7b-hf model and vary layers $l \in$ {$10, 15, 20, 30$} and $\eta \in$ {$0.01, 0.1, 0.5, 1.0, 2.0$}. Results below validate our choice of finding the optimal layer from {15, 30} and fixing $\eta = 1$ (see lines 309-310 in the old draft).
>
> **Sensitivity with respect to layer**
> | | $\eta$  |   coais |   corr |   hallu |   mr |   ref |   surv |   syco |
> |:--|:--|:--|:--|:--|:--|:--|:--|:--|
> | COLD-FD | 10 | 0.52 |   0.58 |0.58 |0.3  |  0.54 |    0.4  |     0.52 |
> | | 15 | 0.48 |   0.46 |0.42 |0.48 |  0.52 |    0.7  |     0.56 |
> | | 20 | 0.48 |   0.44 |0.78 |0.52 |  0.58 |    0.74 |     0.66 |
> | | 30 | 0.48 |   0.42 |0.72 |0.5  |  0.52 |    0.74 |     0.48 |
> | COLD-Kernel | 10 | 0.52 |   0.58 |0.66 |0.48 |  0.38 |    0.72 |     0.52 |
> | | 15 | 0.52 |   0.58 |0.68 |0.48 |  0.38 |    0.72 |     0.52 |
> | | 20 | 0.52 |   0.58 |0.68 |0.48 |  0.38 |    0.72 |     0.52 |
> | | 30 | 0.52 |   0.58 |0.68 |0.48 |  0.38 |    0.72 |     0.52 |
>
>
>
> **Sensitivity with respect to $\eta$**
> | | $\eta$  |   coais |   corr |   hallu |   mr |   ref |   surv |   syco |
> |:--|:--|:--|:--|:--|:--|:--|:--|:--|
> | COLD-FD | 0.01   | 0.46 |   0.58 |0.62 |0.48 |  0.36 |    0.72 |     0.54 |
> | | 0.1 | 0.5  |   0.5  |0.54 |0.56 |  0.48 |    0.68 |     0.56 |
> | | 0.5 | 0.58 |   0.46 |0.48 |0.54 |  0.48 |    0.68 |     0.58 |
> | | 1.0 | 0.52 |   0.64 |0.78 |0.52 |  0.58 |    0.74 |     0.68 |
> | | 2.0 | 0.6  | 0.46 | 0.5  | 0.58 | 0.46 | 0.7  | 0.58 |
> | COLD-Kernel | 0.01 | 0.5  |   0.58 |0.52 |0.48 |  0.38 |    0.56 |     0.42 |
> | | 0.1 | 0.5  |   0.58 |0.52 |0.48 |  0.38 |    0.56 |     0.42 |
> | | 0.5 | 0.5  |   0.58 |0.52 |0.48 |  0.38 |    0.56 |     0.42 |
> | | 1.0 | 0.52 |   0.64 |0.68 |0.48 |  0.38 |    0.72 |     0.52 |
> | | 2.0 | 0.5  | 0.58 | 0.56 | 0.48 | 0.38 | 0.58 | 0.42 |

---

> > ### Author Response · Authors · 2025-11-21
> > **Author Response to Reviewer QoRx (2/3)**
> >
> > > Most results are on small/medium LLMs (7B). No evidence the method scales to larger scale-level or multi-modal models.
> >
> > We have now added Qwen2.5-7B-Instruct as an additional model to demonstrate that COLD can generalize to different LLM families, now spanning over 4 families: Meta’s Llama-2-7B, Google’s Gemma-2-9B, MistralAI’s Mistral-7B-v0.1, and Qwen’s Qwen-2.5-7B.
> >
> > |  |   coais |   |  corr |  |   hallu |   |   mr |  |   ref |  |   surv | |   syco | |
> > |:--|:--|:--|:--|:--|:--|:--|:--|:--|:--|:--|:--|:--|:--|:--|
> > | | pair | pos | pair | pos | pair | pos | pair | pos | pair | pos | pair | pos | pair | pos |
> > | Base |           0.02 |           0.02 | 0.38 | 0.38 |    0.32 |    0.32 |    0.56 |    0.56 |     0.90 |     0.90 |        0.38 |        0.38 | 0.90 | **0.90** |
> > | DiffMean   |   0.02 | -    |     0.48 |   -    |     0.36 |   -    |     0.66 |   -    |   0.90 | -    |0.40 |       -    |  0.92 |-    |
> > | ReFT(vector)          |           0.02 |           0.02 | 0.60 | 0.46 |    0.38 |    0.38 |    0.68 |    0.58 |     0.90 |     0.90 |        0.48 |        0.46 | 0.92 | 0.90 |
> > | COLD-FD       |           **0.98** |           **0.98** | **0.98** | **0.94** |    **0.94** |    **0.78** |    **0.94** |    **0.66** |     0.94 |     0.82 |        **0.76** |        **0.80** | **0.94** | 0.88 |
> > | COLD-Kernel     |           0.02 |           0.02 | 0.38 | 0.38 |    0.32 |    0.34 |    0.56 |    0.58 |     **0.90** |     **0.90** |        0.38 |        0.40 | 0.90 | **0.90** |
> >
> > As we demonstrate with above results,  the cross-architecture robustness suggests that the method is not tightly coupled to a particular model instance. Our primary goal in this work is to introduce and carefully evaluate a new steering framework rather than to perform an exhaustive scaling study, which is mainly constrained by limited computational resources. That said, technically, COLD is model-agnostic and relies only on operations that scale in a predictable way with model size: (i) computing gradients of a scalar loss with respect to either intermediate activations (COLD-Kernel) or parameters (COLD-FD), and (ii) applying a linear intervention to hidden states at a chosen layer.
> >
> > The per-example overhead for COLD-FD at inference time is simply one additional forward pass (to obtain Z(x;θ′)  along with Z(x;θ)), independent of the number of steering examples N; COLD-Kernel incurs a cheap dot-product accumulation over stored activation-gradients. Thus, while the absolute cost increases with model size as usual, the relative overhead compared to standard inference is nearly constant and does not introduce any new scaling bottleneck. We will clarify these scaling properties more explicitly in the revision.
> >
> > Regarding multi-modal models, our work is explicitly scoped to text-based behavior steering and we do not claim multi-modal results. However, the COLD construction only requires (i) a differentiable model, (ii) access to intermediate representations, and (iii) a task loss; these ingredients are also present in vision–language and other multi-modal architectures. In principle, one can define steering losses on text tokens, image features, or joint representations and apply the same one-step gradient-based construction. We view extending COLD to multi-modal settings as a promising direction for future work, but one that is orthogonal to the core contribution of this paper.
> >
> >
> > > COLD-Steer relies on in-context examples to approximate the “one-step learning dynamics.” This inherently depends on the number and quality of examples that can fit into the context window (ICL window). Current LLMs (e.g., Llama-2-7B-chat) have a limited token context.
> >
> > We would like to clarify a potential misunderstanding since we do not use the context window of the LLM space at all, but rather use the in-context examples to obtain a steering vector in the activation space to obtain a similar effect of in-context learning (similar to [3]). Given a set of in-context examples illustrating the desired behavior, we first approximate the effect of updating a specific layer’s parameters in the direction of the gradient that minimizes the loss over these examples. This allows us to estimate how the layer’s activations would change if the model were explicitly trained on the in-context examples. Then, for a new prompt, we steer the LLM’s generation toward the target behavior by applying this estimated activation change directly to the layer. In this way, COLD leverages the learning dynamics of the in-context examples within the activation space rather than the prompt space. Thus, it is not constrained by the LLM’s context window, since the examples do not occupy space in the prompt.

---

> > > ### Author Response · Authors · 2025-11-21
> > > **Author Response to Reviewer QoRx (3/3)**
> > >
> > > > The authors acknowledge this limitation briefly (“future work should develop more sophisticated approximations of the neural tangent kernel”) but provide no empirical study on how κ or layer l affect the approximation quality.
> > >
> > > We would like to point the reviewer to Table 10 (Appendix), where we go beyond a simple unit kernel and use the constant and random projection methods as the kernel approximations. As described in lines 193-194 (old draft), the constant kernel approximates similarity by taking the inner product between activations, whereas the random projection kernel computes the inner product after projecting the activations onto a random vector space. However, we find that despite its simplicity, the unit kernel is often the most stable across behaviors. This leaves room for improvement in formulating a more effective approximation of the empirical neural tangent kernel, as also highlighted in the future directions in Section 5. Although our work demonstrates that simple approximations can outperform current baselines, we believe a deeper investigation into more sophisticated, theoretically grounded approximations of the empirical neural tangent kernel is beyond the scope of this paper and merits a separate study. We also provide an empirical analysis on different layers in our response above on layer-wise ablation.
> > >
> > > [1] Park, Kiho, Yo Joong Choe, and Victor Veitch. "The linear representation hypothesis and the geometry of large language models." ICML 2024.
> > >
> > > [2] Arditi, Andy, et al. "Refusal in language models is mediated by a single direction." NeurIPS 2024.
> > >
> > > [3] Liu, Sheng, et al. "In-context vectors: Making in context learning more effective and controllable through latent space steering." ICML 2024.

---

### Official Review · Reviewer_wgtt · 2025-10-31

**Soundness:** 3
**Presentation:** 2
**Contribution:** 3
**Rating:** 6
**Confidence:** 2

**Summary:**

The paper presents COLD-Steer, a training-free and sample-efficient method for steering large language models at inference time. It approximates how model representations would change after a single gradient update on a few in-context examples, enabling behavioral control without retraining. Two variants are proposed: COLD-Kernel-Steer, which aggregates gradient signals using a simple kernel, and COLD-FD-Steer, which uses a finite-difference approximation requiring two forward passes. Experiments on CAA, BiPO, and OpinionsQA show that COLD-Steer achieves similar or better control than prior methods with 10–50 times fewer examples.

**Strengths:**

- Interesting idea of approximating learning dynamics to perform activation steering.
- Training-free and efficient compared to fine-tuning or parameter-tuning approaches.
- Works with few examples and across different LLM families.
- Strong empirical results on several behavioral control tasks.
- Both variants are complementary, with COLD-FD providing more consistent results than COLD-Kernel, though at the expense of computational efficiency.

**Weaknesses:**

- Theoretical justification of approximations (unit kernel, finite difference) is limited.
- While examples of COLD-steered generations are given and discussed, the paper could benefit from more interpretability analysis of how activations are actually changed.

**Questions:**

- How sensitive is COLD-Steer to the choice of steering layer and the η multiplier?
- Could kernel approximations beyond the unit kernel improve stability without major cost?
- COLD-FD reduces memory use by clipping small parameter updates, keeping only about 4% of parameters with significant changes. Could you provide more detail on how the clipping $\theta_\text{thresh}$ threshold is chosen and how it affects steering performance?

---

> ### Author Response · Authors · 2025-11-21
> **Author Response to Reviewer wgtt (1/2)**
>
> We are delighted by the reviewer’s positive support of our work as they find our approach interesting, complementary and efficient, and our experiments comprehensive. They also raise very valuable concerns that help us improve parts of the exposition. We provide our responses to those concerns below and will also revise our paper to incorporate our discussions.
>
> > Theoretical justification of approximations (unit kernel, finite difference) is limited.
>
> We note that the finite difference approximation is theoretically justified in the definition of the gradient and the Taylor Series approximation of Equation 3. However, we thank the reviewer for giving us an opportunity to revisit our justification behind the effectiveness of the unit kernel approximation. In particular, we investigate why a unit kernel can be effective in general. Specifically, we observe that the approximation $\langle \nabla_{\theta} Z(x_i), \nabla_{\theta} Z(x_j) \rangle \approx 1$ can hold when the parameter gradients of a subnetwork are approximately the same (up to scaling) across inputs in a dataset. This occurs when the per-example gradient vectors are highly aligned or dominated by a single common direction. In such cases, each entry is roughly equal to the product of two similar norms, and after normalization, the resulting kernel closely resembles a unit (all-ones) kernel. Since all the inputs in the dataset are designed to elicit the same underlying behavior, we can expect the gradients with respect to the model’s parameters to be highly aligned. This is based on the assumption that the model internally encodes the relevant high-level concept (such as specific behaviors) in a consistent and linear way across inputs, which is often called the linear representation hypothesis [1,2]. In other words, the directions in activation or gradient space corresponding to a particular concept are similar across different inputs that express the concept. This explains why the kernel, computed as the inner product of per-example gradients, can be well-approximated by a unit (all-ones) matrix: each input contributes a gradient pointing along the same underlying conceptual direction, making them appear nearly identical in the kernel space after normalization.
>
> > While examples of COLD-steered generations are given and discussed, the paper could benefit from more interpretability analysis of how activations are actually changed.
>
> We thank the reviewer for this suggestion. Our work is in fact strongly motivated by mechanistic interpretability: COLD constructs explicit, loss-derived steering directions in activation space that encode how the model would learn a given behavior in one gradient step. This provides a compact, mechanistically meaningful vector (or small set of vectors) at a specific layer, that captures how the model’s internal representations should be adjusted to elicit a target behavior.
>
> We agree that a deeper interpretability analysis of these steering directions is an exciting next step. Because COLD operates directly on intermediate activations, it naturally lends itself to mechanistic tools such as: Decomposing the steering vector into contributions from specific attention heads or MLP neurons; Activation patching, where we selectively replace the activations of particular tokens or layers across runs to localize where the behavior is implemented, and Low-dimensional projections or feature visualization, including 2D projections similar to contemporaneous work [3], to visualize the effect of steering vectors such as COLD.
>
> In this paper, we focused on establishing the behavioral and methodological contributions of COLD, and a full mechanistic analysis of the learned steering directions is beyond scope. However, we view one of the main advantages of our framework as precisely that it produces structured, gradient-grounded activation directions that are well-suited for future mechanistic interpretability studies. We will clarify this connection, and note concrete interpretability avenues (e.g., projections and activation patching), in the revised version.

---

> > ### Author Response · Authors · 2025-11-21
> > **Author Response to Reviewer wgtt (2/2)**
> >
> > > How sensitive is COLD-Steer to the choice of steering layer and the η multiplier?
> >
> > We analyze the sensitivity of COLD-Steer with respect to the choice of steering layer and $\eta$. In particular, we consider the Llama-2-7b-hf model and vary layers $l \in$ {$10, 15, 20, 30$} and $\eta \in$ {$0.01, 0.1, 0.5, 1.0, 2.0$}. Results below validate our choice of finding the optimal layer from {15, 30} and fixing $\eta = 1$ (see lines 309-310 in the old draft).
> >
> > **Sensitivity with respect to layer**
> > | | layer  |   coais |   corr |   hallu |   mr |   ref |   surv |   syco |
> > |:--|:--|:--|:--|:--|:--|:--|:--|:--|
> > | COLD-FD | 10 | 0.52 |   0.58 |0.58 |0.3  |  0.54 |    0.4  |     0.52 |
> > | | 15 | 0.48 |   0.46 |0.42 |0.48 |  0.52 |    0.7  |     0.56 |
> > | | 20 | 0.48 |   0.44 |0.78 |0.52 |  0.58 |    0.74 |     0.66 |
> > | | 30 | 0.48 |   0.42 |0.72 |0.5  |  0.52 |    0.74 |     0.48 |
> > | COLD-Kernel | 10 | 0.52 |   0.58 |0.66 |0.48 |  0.38 |    0.72 |     0.52 |
> > | | 15 | 0.52 |   0.58 |0.68 |0.48 |  0.38 |    0.72 |     0.52 |
> > | | 20 | 0.52 |   0.58 |0.68 |0.48 |  0.38 |    0.72 |     0.52 |
> > | | 30 | 0.52 |   0.58 |0.68 |0.48 |  0.38 |    0.72 |     0.52 |
> >
> >
> >
> > **Sensitivity with respect to $\eta$**
> > | | $\eta$  |   coais |   corr |   hallu |   mr |   ref |   surv |   syco |
> > |:--|:--|:--|:--|:--|:--|:--|:--|:--|
> > | COLD-FD | 0.01   | 0.46 |   0.58 |0.62 |0.48 |  0.36 |    0.72 |     0.54 |
> > | | 0.1 | 0.5  |   0.5  |0.54 |0.56 |  0.48 |    0.68 |     0.56 |
> > | | 0.5 | 0.58 |   0.46 |0.48 |0.54 |  0.48 |    0.68 |     0.58 |
> > | | 1.0 | 0.52 |   0.64 |0.78 |0.52 |  0.58 |    0.74 |     0.68 |
> > | | 2.0 | 0.6  | 0.46 | 0.5  | 0.58 | 0.46 | 0.7  | 0.58 |
> > | COLD-Kernel | 0.01 | 0.5  |   0.58 |0.52 |0.48 |  0.38 |    0.56 |     0.42 |
> > | | 0.1 | 0.5  |   0.58 |0.52 |0.48 |  0.38 |    0.56 |     0.42 |
> > | | 0.5 | 0.5  |   0.58 |0.52 |0.48 |  0.38 |    0.56 |     0.42 |
> > | | 1.0 | 0.52 |   0.64 |0.68 |0.48 |  0.38 |    0.72 |     0.52 |
> > | | 2.0 | 0.5  | 0.58 | 0.56 | 0.48 | 0.38 | 0.58 | 0.42 |
> >
> >
> > > Could kernel approximations beyond the unit kernel improve stability without major cost?
> >
> > We agree that this is a valuable question that we have explored in Table 10 (Appendix). As described in lines 193-194 (old draft), the constant kernel approximates similarity by taking the inner product between activations, whereas the random projection kernel computes the inner product after projecting the activations onto a random vector space. In particular, we go beyond a simple unit kernel and use the constant and random projection methods as the kernel approximations. However, we find that despite its simplicity, unit kernel is often the most stable across behaviors. This leaves room for improvement in formulating a more effective approximation of the empirical neural tangent kernel, as also highlighted in the future directions in Section 5. Although our work demonstrates that simple approximations can outperform current baselines, we believe a deeper investigation into more sophisticated, theoretically grounded approximations of the empirical neural tangent kernel is beyond the scope of this paper and merits a separate study.
> >
> > > COLD-FD reduces memory use by clipping small parameter updates, keeping only about 4% of parameters with significant changes. Could you provide more detail on how the clipping  threshold is chosen and how it affects steering performance?
> >
> > We would like to apologize for the misunderstanding caused by that statement. We would like to clarify that we do not perform any parameter clipping to save memory, as it does not help in our current implementation since PyTorch functional abstraction requires passing the full parameter set instead of a subset. Instead, we wanted to provide that observation to motivate future more space-efficient implementations of COLD-FD. However, we provide some preliminary results to show how the clipping threshold affects the number of parameters and the performance. Developers can thus trade off the memory complexity for the performance by tuning this clipping threshold, but since this is not the focus of our work, we will defer it to the Appendix and clarify the previous statement.
> >
> >
> > |Threshold | Accuracy | Number of parameters |
> > |:--|:--|:--|
> > | 0 | 0.72 | 3.14e+9 |
> > | 1e-12 | 0.68 | 2.12e+9 |
> > | 1e-10 | 0.64 | 5.28e+6 |
> > | 1e-9 | 0.6 | 43k |
> > | 1e-8 | 0.6 | 1267 |
> > ​​
> >
> >
> > [1] Park, Kiho, Yo Joong Choe, and Victor Veitch. "The linear representation hypothesis and the geometry of large language models." ICML 2024.
> >
> >
> > [2] Arditi, Andy, et al. "Refusal in language models is mediated by a single direction." NeurIPS 2024.
> >
> > [3] Vu, Hieu M., and Tan M. Nguyen. "Angular steering: Behavior control via rotation in activation space." NeurIPS 2025.

---

> > > ### Comment · Reviewer_wgtt · 2025-11-26
> > >
> > > I thank the authors for their clarifications, which have effectively addressed my concerns. I am satisfied with the answers provided, and I will therefore keep my original score. Overall, I believe the paper is above the acceptance threshold.

---

> > > > ### Author Response · Authors · 2025-11-26
> > > >
> > > > Thank you for your prompt response! We are pleased that our comments have addressed your concerns, and we remain happy to clarify any additional points that may arise during the discussion period.

---

### Official Review · Reviewer_toPp · 2025-11-01

**Soundness:** 2
**Presentation:** 3
**Contribution:** 3
**Rating:** 4
**Confidence:** 3

**Summary:**

The paper proposes COLD-Steer, a framework for steering large language model (LLM) activations by approximating the representational changes that would result from gradient descent on in-context examples. It introduces two variants:
- COLD-Kernel Steer: Uses kernel approximation to estimate gradients.
- COLD-FD Steer: Employs finite-difference approximation to estimate gradients.
The approach is claimed to be training-free, data-efficient, and unifying existing contrastive steering methods. Additionally, it is presented as being applicable across a diverse set of steering tasks.

**Strengths:**

- Strong theoretical motivation and a unifying perspective that generalizes existing methods. It would be valuable to further elaborate on the connections to other approaches such as [1,2,3].
- Includes computational complexity analysis, but a more explicit comparison with the complexity of existing methods would strengthen the contribution.
- Extensive experimental setup, covering selection and open-generation tasks, distribution shifts, computational efficiency, and qualitative outputs.
- Compares against a broad range of baselines, demonstrating the method’s effectiveness across diverse scenarios.

[1] Refusal in Language Models Is Mediated by a Single Direction
[2] Controlling Language and Diffusion Models by Transporting Activations
[3] Angular Steering: Behavior Control via Rotation in Activation Space

**Weaknesses:**

See Questions

**Questions:**

- Line 40: claims existing methods use between 250 to 1000 examples, but [1] uses as few as 64. This counterexample should be addressed.
- Figure 1 (left): Why does the contrastive method significantly decrease in accuracy as the number of samples increases? Which experiments demonstrate this phenomenon?
- Lines 191–193: claims that using a unit kernel yields strong empirical performance. Some discussion to explain this observation would be helpful.
- Table 2: COLD-FD performs much better than ReFT(mlp), which is a more complex method. What explains this? COLD-FD only approximates the gradient, whereas ReFT performs actual gradient descent.
- Table 2 (top) and Table 5: COLD-Kernel is a generalization of DiffMean/CAA ([2]), but performs worse. Why is this the case?
- Table 2 should include an average (avg) column for easier comparison across methods.
- More evaluation on different LLM families and sizes is needed. Table 3 only shows results for Gemma 2 9B and Mistral 7B on the selection task. Other tasks lack cross-model comparison. Including a more diverse set of sizes would better demonstrate generalization.
- Lines 306–307: "Steering is applied to all prompt token representations (rather than the final token only), which yields consistently better performance." Does this mean steering is applied to the input token representations? If so, is it applied sequentially during steering vector computation as in [3]? Does this setup apply to both selection and open-ended tasks?
- Line 309: is η the same across all methods? Do all methods perform best at η = 1? Please provide full grid search results for all methods.
- Lines 311–312: "For open-ended generation, we intervene only at the first generated token to guide continuation, while limiting the compounding effects of steering." Is this strategy used for the proposed methods only or all baselines? Many existing methods steer on all generated tokens [1,2,3,4]. An empirical comparison would help justify this decision.
- Table 4: lacks comparison to other methods. Please include baseline results.
- The paper lacks robustness evaluation: does the method inadvertently affect untargeted behaviors while steering?

References:
[1] Refusal in Language Models Is Mediated by a Single Direction
[2] Steering Llama 2 via Contrastive Activation Addition
[3] Controlling Language and Diffusion Models by Transporting Activations
[4] Angular Steering: Behavior Control via Rotation in Activation Space

---

> ### Author Response · Authors · 2025-11-21
> **Author Response to Reviewer toPp (1/5)**
>
> We thank the reviewer for their detailed feedback and are glad that they find our approach theoretically well motivated, unifying, and experimental comparison thorough. However, they have also raised some concerns that we address below in our rebuttal. We kindly invite the reviewer to engage in further discussion should any points remain unclear or require additional elaboration.
>
>
> > It would be valuable to further elaborate on the connections to other approaches such as [1,3,4].
>
> We thank the reviewer for pointing us to these references. [1] shows that refusal behavior in language models can be mediated by a single direction across layers. They extract this direction by leveraging the difference of means (DiffMean) method on a single layer, which we already consider as a baseline. However, instead of adding this direction to that layer during inference, they remove this “behavior vector” from all layers as they focus on a different problem of directional ablation than activation addition for behavioral steering. While our work is concerned with identifying the optimal activation addition for a given layer, [3] tackles the complementary problem of layer search in current methods. Their approach extends a simple greedy layer search by applying optimal transport maps sequentially across all layers. [4] provides a novel but contemporaneous perspective (since the code and arXiv were made available only on 30th October 2025) to rotate activations in a 2D space spanned by the DiffMean vector of one layer and the PCA component of all layers. Since we form a different gradient-inspired steering direction, we believe that more gains can be achieved by combining angular steering using COLD vectors, but we leave this as future work, as it is a parallel advancement.
>
> We also incorporate this discussion in the revised draft by extending our discussion on activation steering (in Appendix A), and will include it in the main paper with the additional page upon acceptance.
>
>
> > Complexity analysis comparison with baselines
>
> We would like to point the reviewer to Section 3.3 and Table 1, where we have already explicitly compared the complexities of the representative baselines (Contrastive (DiffMean-like) and Parameter-tuning (ReFT-like)). Here, we separate the time and space complexities needed to incorporate the behavior from given examples and use it to intervene on a new example. In addition, the reviewer’s pointed references are either contemporaneous [4] or are not directly applicable [1,3] to our problem of in-context behavioral steering. Thus, we avoid a comparison of complexity analysis against them. If the reviewer thinks the discussion is still insufficient, we would be happy to further expand it based on the reviewer’s guidance.
>
>
> > Line 40: claims existing methods use between 250 to 1000 examples, but [1] uses as few as 64. This counterexample should be addressed.
>
>
> We would like to clarify that we made this statement in the context of how well humans are able to “grasp such behavioral shifts from just a handful of cases”. Thus, we mean that existing baselines (such as ReFT) need a high number of examples to reach a high performance (such as 0.90 in Figure 1, right). Note that DiffMean or the contrastive baseline in Figure 1, right (which is also used by [1]), shows similar performance for fewer examples, but it is not able to reach a high enough human-like performance. We would clarify this further in a footnote there.
>
>
> > Figure 1 (left): Why does the contrastive method significantly decrease in accuracy as the number of samples increases? Which experiments demonstrate this phenomenon?
>
>
> This is also highlighted by experiments in Figure 3 or Figure 4, where we show the effect of the number of examples on the performance. As shown, this trend can be dataset-specific, and the accuracy of the contrastive method can go down since a simple mean aggregation can be sensitive to outliers in the dataset, and their chance of occurring increases when we increase the number of samples.

---

> ### Author Response · Authors · 2025-11-21
> **Author Response to Reviewer toPp (2/5)**
>
> > Lines 191–193: claims that using a unit kernel yields strong empirical performance. Some discussion to explain this observation would be helpful.
>
> We thank the reviewer for this question, as it gives us an opportunity to investigate why a unit kernel can be effective in general. Specifically, we observe that the approximation $\langle \nabla_{\theta} Z(x_i), \nabla_{\theta} Z(x_j) \rangle \approx 1$ can hold when the parameter gradients of a subnetwork are approximately the same (up to scaling) across inputs in a dataset. This occurs when the per-example gradient vectors are highly aligned or dominated by a single common direction. In such cases, each entry is roughly equal to the product of two similar norms, and after normalization, the resulting kernel closely resembles a unit (all-ones) kernel. Since all the inputs in the dataset are designed to elicit the same underlying behavior, we can expect the gradients with respect to the model’s parameters to be highly aligned. This is based on the assumption that the model internally encodes the relevant high-level concept (such as specific behaviors) in a consistent and linear way across inputs, which is often called the linear representation hypothesis [1,5]. In other words, the directions in activation or gradient space corresponding to a particular concept are similar across different inputs that express the concept. This explains why the kernel, computed as the inner product of per-example gradients, can be well-approximated by a unit (all-ones) matrix: each input contributes a gradient pointing along the same underlying conceptual direction, making them appear nearly identical in the kernel space after normalization.
>
>
> > Table 2: COLD-FD performs much better than ReFT(mlp), which is a more complex method. What explains this? COLD-FD only approximates the gradient, whereas ReFT performs actual gradient descent.
>
> While ReFT actually performs the gradient descent, its effectiveness is limited as it uses it to train a set of parameters on top of an activation, which is sensitive to the number of training examples due to overfitting issues. In contrast, our approach is training-free and leverages the gradient to directly obtain the steering vector, building on the theoretical framework from [6]. Note that when we have a large number of examples, ReFT can train effective parameters to outperform COLD-FD (see Figure 1, right), but suffers in the low-sample scenario, which is the focus of this work.
>
>
> > Table 2 (top) and Table 5: COLD-Kernel is a generalization of DiffMean/CAA ([2]), but performs worse. Why is this the case?
>
> This is because of the difference in the choice of the loss function between the two. Note that the only difference between DiffMean and COLD-Kernel (with the unit kernel) in the pair setting is that COLD-Kernel uses the DPO loss whereas DiffMean uses the negative l2 distance as the loss function (due to Corollary 1). It turns out that DPO loss is not as effective as the negative l2 distance loss for the pairwise settings. On the other hand, we find that when using SFT loss (in the positive-only behavior setting), COLD-Kernel performs better than even DiffMean. This is an interesting result which highlights the intrinsic effect of the gradient vector of different losses within LLMs. Given that the focus of our work is to demonstrate the effectiveness of using the one-step gradient vector for steering, we leave a deeper theoretical analysis of how specific loss gradients influence behavior to future works.
>
> > Table 2 should include an average (avg) column for easier comparison across methods.
>
> We thank the reviewer for this suggestion, but since certain behaviors may have higher general accuracies than others, a simple average may not be an appropriate and robust metric to such outliers. Thus, we have instead included an average rank for each method across different behaviors. To compute average rank, we first rank the performance of all methods within each behavior (1 = best, 2 = second best, etc.). We then average these ranks across behaviors. This metric reduces the influence of behaviors with unusually high or low absolute accuracies and offers a fairer comparison across different methods. As shown below (lower is better), COLD-FD and COLD-Kernel rank the highest on average compared to other methods.
>
> **Average rank for Llama-2-7b-chat-hf in Table 2**
> | Method | pair | pos |
> |:--|:--|:--|
> | Base | 5.14 | 4.43 |
> | Base (ICL) | 7.14 | 4.29 |
> | DiffMean | 4.00 | - |
> | ICV | 5.29 | - |
> | DiffMeanPW | 4.57 | - |
> | DiffMeanProj | 4.71 | - |
> | ReFT (mlp) | 5.29 | 4.00 |
> | ReFT (vector) | 3.29 | 3.14 |
> | COLD-FD | **2.00** | **1.71** |
> | COLD-Kernel | 4.43 | 2.57 |
>
> **Average rank for Llama-2-7b-hf in Table 2**
> | Method | pair | pos |
> |:--|:--|:--|
> | Base | 2.00 | 2.43 |
> | Base(ICL) | 2.71 | 2.86 |
> | DiffMean | 4.43 | - |
> | ReFT(mlp) | 5.14 | 4.14 |
> | ReFT(vector) | 2.86 | 4.43 |
> | COLD-FD | **1.29** | 2.00 |
> | COLD-Kernel | 2.43 | **1.57** |

---

> ### Author Response · Authors · 2025-11-21
> **Author Response to Reviewer toPp (3/5)**
>
> > More evaluation on different LLM families and sizes is needed. Table 3 only shows results for Gemma 2 9B and Mistral 7B on the selection task. Other tasks lack cross-model comparison. Including a more diverse set of sizes would better demonstrate generalization.
>
> We have added Qwen2.5-7B-Instruct as an additional model to demonstrate that COLD can generalize to different LLM families, thus expanding our analysis over 4 different families: Meta’s Llama-2-7B, Google’s Gemma-2-9B, MistralAI’s Mistral-7B-v0.1, and Qwen’s Qwen-2.5-7B-Instruct.
>
> **Complete selection task for Qwen-2.5-7B-Instruct**
> |  |   coais |   |  corr |  |   hallu |   |   mr |  |   ref |  |   surv | |   syco | |
> |:--|:--|:--|:--|:--|:--|:--|:--|:--|:--|:--|:--|:--|:--|:--|
> | | pair | pos | pair | pos | pair | pos | pair | pos | pair | pos | pair | pos | pair | pos |
> | Base |           0.02 |           0.02 | 0.38 | 0.38 |    0.32 |    0.32 |    0.56 |    0.56 |     0.90 |     0.90 |        0.38 |        0.38 | 0.90 | **0.90** |
> | DiffMean   |   0.02 | -    |     0.48 |   -    |     0.36 |   -    |     0.66 |   -    |   0.90 | -    |0.40 |       -    |  0.92 |-    |
> | ReFT(vector)          |           0.02 |           0.02 | 0.60 | 0.46 |    0.38 |    0.38 |    0.68 |    0.58 |     0.90 |     0.90 |        0.48 |        0.46 | 0.92 | 0.90 |
> | COLD-FD       |           **0.98** |           **0.98** | **0.98** | **0.94** |    **0.94** |    **0.78** |    **0.94** |    **0.66** |     0.94 |     0.82 |        **0.76** |        **0.80** | **0.94** | 0.88 |
> | COLD-Kernel     |           0.02 |           0.02 | 0.38 | 0.38 |    0.32 |    0.34 |    0.56 |    0.58 |     0.90 |     0.90 |        0.38 |        0.40 | 0.90 | 0.90 |
>
>
> In addition, we also provide results for the generation task for Mistral and Qwen models below:
>
> **CAA Generation task for Qwen-2.5-7B-Instruct**
> |  |   coais |   corr |   hallu |   mr |  ref |  surv |  syco |
> |:--|:--|:--|:--|:--|:--|:--|:--|
> | Base | **0.34** |   6.54 | 0.78 |1.38 |  **3.86** |    **7.48** | 0.72 |
> | DiffMean |  0.2  |   6.84   | 1.06  |1.38    |   3.44 |    7.12    | 0.72 |
> | ReFT(vector)  | 0.14    |   **6.96** | 0.94 | 1.52   |   3.48  |    7.04 | **0.85** |
> | COLD-FD   | 0.16     |   2.28  | **9.98** | 2.34    |   4.9   |    5.76       | 0.83 |
> | COLD-Kernel  | 0.26   |   6.3   | 0.58 | 1.66  | 3.72  |    7.24    | 0.69 |
>
> **CAA Generation task for Mistral-7B-v0.1**
> | |   coais |   corr |   hallu |   mr |  ref |  surv |  syco |
> |:--|:--|:--|:--|:--|:--|:--|:--|
> | Base | 0.48  | 6.08  |3.74 |2.14 |  1.1  |   7.66 | 1.11 |
> | Contrastive(linear) | 3    |   7.76 | 4.02 | 2   | 1.96  |    **7.82**    | 1.26 |
> | ReFT(vector)   | 0.66    |   6.66    |3.92    | 2.42    |     1.56  |    7.76    | 1.15 |
> | COLD-FD   | **4.64**   |   **8.52** | **8.52** | **2.88** | **7.54**  |    7.38 | **1.47**  |
> | COLD-Kernel | 0.4    |   6.24 | 3.76    | 2.38    | 1.54    |    7.66  | 1.06  |
>
> As we demonstrate with above results,  the cross-architecture robustness suggests that the method is not tightly coupled to a particular model instance. Our primary goal in this work is to introduce and carefully evaluate a new steering framework rather than to perform an exhaustive scaling study, which is mainly constrained by limited computational resources. That said, technically, COLD is model-agnostic and relies only on operations that scale in a predictable way with model size: (i) computing gradients of a scalar loss with respect to either intermediate activations (COLD-Kernel) or parameters (COLD-FD), and (ii) applying a linear intervention to hidden states at a chosen layer.
>
> The per-example overhead for COLD-FD at inference time is simply one additional forward pass (to obtain Z(x;θ′)  along with Z(x;θ)), independent of the number of steering examples N; COLD-Kernel incurs a cheap dot-product accumulation over stored activation-gradients. Thus, while the absolute cost increases with model size as usual, the relative overhead compared to standard inference is nearly constant and does not introduce any new scaling bottleneck. We will clarify these scaling properties more explicitly in the revision.
>
>
> > Lines 306–307: "Steering is applied to all prompt token representations (rather than the final token only), which yields consistently better performance." Does this mean steering is applied to the input token representations? If so, is it applied sequentially during steering vector computation as in [3]? Does this setup apply to both selection and open-ended tasks?
>
> Yes, we apply mean steering to all input token representations, but unlike [3], we do not intervene on multiple layers of the model and rather intervene on a single most effective layer following [2]. Thus, we do not apply sequential vector computation but instead calculate the steering vector given the prompt and add it to all prompt token representations at a layer in parallel. We follow this setup for all tasks.

---

> ### Author Response · Authors · 2025-11-21
> **Author Response to Reviewer toPp (4/5)**
>
> > Line 309: is η the same across all methods? Do all methods perform best at η = 1? Please provide full grid search results for all methods.
>
> Yes, we keep $\eta = 1$ for all methods for a fair comparison when the same strength of steering is applied. Furthermore, we apply normalization to contrastive methods, following existing works [2, 6], while parameter-tuning methods do not have $\eta$ as the scale can be learned in the parameter space. For completeness, we show below the effect of steering the Llama-2-7b-hf model for applicable methods across multiple steering strengths. We can note that $\eta=1$ performs consistently better than other strengths, validating our final choice.
>
>
> | | $\eta$  |   coais |   corr |   hallu |   mr |   ref |   surv |   syco |
> |:--|:--|--:|--:|--:|--:|--:|--:|--:|
> | DiffMean | 0.01 | 0.52 |  0.58 |  0.68 |  0.48 |   0.36 | 0.72 |  0.52 |
> | | 0.1 | 0.52 | 0.58 | 0.68 | 0.48 | 0.36 | 0.72 | 0.52 |
> | | 0.5 | 0.54 | 0.58 | 0.7  | 0.48 | 0.36 | 0.72 | 0.54 |
> | | 1.0 | 0.58 | 0.62 | 0.7  | 0.48 | 0.38 | 0.72 | 0.54 |
> | | 2.0 | 0.56 | 0.58 | 0.68 | 0.5  | 0.36 | 0.72 | 0.56 |
> | COLD-FD | 0.01   | 0.46 |   0.58 |0.62 |0.48 |  0.36 |    0.72 |     0.54 |
> | | 0.1 | 0.5  |   0.5  |0.54 |0.56 |  0.48 |    0.68 |     0.56 |
> | | 0.5 | 0.58 |   0.46 |0.48 |0.54 |  0.48 |    0.68 |     0.58 |
> | | 1.0 | 0.52 |   0.64 |0.78 |0.52 |  0.58 |    0.74 |     0.68 |
> | | 2.0 | 0.6  | 0.46 | 0.5  | 0.58 | 0.46 | 0.7  | 0.58 |
> | COLD-Kernel | 0.01 | 0.5  |   0.58 |0.52 |0.48 |  0.38 |    0.56 |     0.42 |
> | | 0.1 | 0.5  |   0.58 |0.52 |0.48 |  0.38 |    0.56 |     0.42 |
> | | 0.5 | 0.5  |   0.58 |0.52 |0.48 |  0.38 |    0.56 |     0.42 |
> | | 1.0 | 0.52 |   0.64 |0.68 |0.48 |  0.38 |    0.72 |     0.52 |
> | | 2.0 | 0.5  | 0.58 | 0.56 | 0.48 | 0.38 | 0.58 | 0.42 |
>
>
>
> > Lines 311–312: "For open-ended generation, we intervene only at the first generated token to guide continuation, while limiting the compounding effects of steering." Is this strategy used for the proposed methods only or all baselines? Many existing methods steer on all generated tokens [1,2,3,4]. An empirical comparison would help justify this decision.
>
> Yes, we use this strategy of only intervening on the prompt for all methods for a fair comparison. This allows us to limit the effects of compounding and reduce the generation time as well. Nevertheless, we provide an empirical analysis of steering over successive generations and find the results for Llama-2-7b-chat-hf shown below. We find that steering on all generated tokens does not consistently increase performance as compared to just steering on the prompt, and in many cases, the performance actually goes down. We believe the reduction in performance arises as small errors in the steering vectors can compound upon applying them on every generated token. Thus, for consistency and efficiency (since steering at every generation can be costly), we follow the setup of steering just the prompt representations (i.e., the first generated token).
>
>
> |  | steer at   |   coais |   corr |   hallu |   mr |   ref |   surv |   syco |
> |:--|:--|--:|--:|--:|--:|--:|--:|--:|
> | COLD-Kernel | prompt-only | **0.20** | 3.86 | **3.30** | **2.22** | 5.20 | 5.40 | **0.96** |
> | COLD-Kernel | all | 0.16 | **4.36** | 3.08 | 2.10 | **5.22** | **5.72** | 0.74 |
> | COLD-FD | prompt-only | **0.82** | **5.06** | 3.32 | 2.62 | 4.92 | **6.20** | **1.23** |
> | COLD-FD | all | 0.6 | 3.96 |  **10** |    **3.02** |   **8.4** | 4.98 |    0.81 |
>
>
>
> > Table 4: lacks comparison to other methods. Please include baseline results.
>
>
> Below, we present the complete results of Table 4 to compare steering methods on the generation task of the CAA. We find that COLD performs consistently well as compared to more baselines in the generation task of the CAA dataset as well.
>
>
> | |   coais |   corr |   hallu |   mr |   ref |   surv |   syco |
> |:--|--:|--:|--:|--:|--:|--:|--:|
> | Llama-2-7b-hf | | | | | | | |
> | Base | 4.30 | 3.80 | 5.98 | 4.84 | 3.16 | **4.84** | **4.32** |
> | DiffMean | **5.33** | 3.08 | 7.2 | 5.02 | 3.64 | 4.76 | 4.15 |
> | ReFT(vector) | 3.92 | 2.36 | 7   | 5.38 | 3.88 | 4.66 | 4.24 |
> | COLD-FD | 3.94 | 2.58 | **7.22** | **5.18** | **4.50** | 4.36 | 4.06 |
> | COLD-Kernel | 4.36 | **3.84** | 6.04 | 4.53 | 2.80 | 4.76 | 3.68 |
> | Llama-2-7b-chat-hf | | | | | | | |
> | Base | 0.28 | 3.82 | 2.98 | 1.98 | 4.88 | 5.26 | 0.92 |
> | DiffMean |0.3 |  4.4 |    2.64 |    2.08 |      5.5 |        6.04 |    0.81 |
> | ReFT(vector) |  0.14 | 4.46 |    2.92 |   **2.66** |   **5.2** |        **6.22** |    0.69 |
> | COLD-FD | **0.82** | **5.06** | **3.32** | 2.62 | 4.92 | 6.20 | **1.23** |
> | COLD-Kernel | 0.20 | 3.86 | 3.30 | 2.22 | **5.20** | 5.40 | 0.96 |

---

> > ### Author Response · Authors · 2025-11-21
> > **Author Response to Reviewer toPp (5/5)**
> >
> > > The paper lacks robustness evaluation: does the method inadvertently affect untargeted behaviors while steering?
> >
> > The focus of this paper is on steering the LLM toward a specific target behavior as desired by the user. In untargeted scenarios (i.e., when there is no targeted behavior desired by the user), steering can be simply turned off to avoid unintended effects. However, as the reviewer may be suggesting, even when steering a specific behavior, it is still possible that steering one behavior may sometimes influence others. For instance, sycophancy might increase when attempting to reduce hallucinations. But current steering benchmarks do not test for such behavioral overlaps, making systematic analysis extremely challenging. While we acknowledge that this is an important question, addressing it would require a separate study, as there are currently no labeled datasets or prior analyses in the literature to support such an investigation.
> >
> >
> > [1] Arditi, Andy, et al. "Refusal in language models is mediated by a single direction." NeurIPS 2024
> >
> >
> > [2] Rimsky, Nina, et al. "Steering llama 2 via contrastive activation addition." ACL 2024.
> >
> >
> > [3] Rodriguez, Pau, et al. "Controlling Language and Diffusion Models by Transporting Activations." ICLR 2025.
> >
> >
> > [4] Vu, Hieu M., and Tan M. Nguyen. "Angular steering: Behavior control via rotation in activation space." NeurIPS 2025.
> >
> > [5] Park, Kiho, Yo Joong Choe, and Victor Veitch. "The linear representation hypothesis and the geometry of large language models." ICML 2024.
> >
> >
> > [6] Yi Ren and Danica J Sutherland. Learning dynamics of llm finetuning. ICLR 2025.
> >
> >
> > [7] Zhengxuan Wu, Aryaman Arora, Atticus Geiger, Zheng Wang, Jing Huang, Dan Jurafsky, Christopher D Manning, and Christopher Potts. Axbench: Steering llms? even simple baselines outperform sparse autoencoders. ICML 2025.

---

### Official Review · Reviewer_xSG1 · 2025-11-02

**Soundness:** 4
**Presentation:** 3
**Contribution:** 3
**Rating:** 8
**Confidence:** 3

**Summary:**

This paper tries to fill the gap between in-context learning and parameter efficient tuning by approximating the change in the representations when finetuning on in-context examples at inference time. They propose two training-free approaches to achieve this goal, a unit kernel approximation method and a finite-difference approximation method. The proposed methods are tested on multiple choice data as well as open-ended generations. They are able to achieve up to 95% performance with 10-50 times fewer examples compared to several steering and parameter efficient tuning approaches. In addition, the proposed approaches do not require pairs of positive and negative examples in contrast to other activation engineering approaches such as Contrastive Activation Addition (CAA).

**Strengths:**

- This paper tackles an important and interesting problem.

- The work is grounded in prior literature and does a good job telling a coherent and concise story.

 - The proposed approaches are theoretically grounded.

 - There are several in-depth experiments that evaluate the proposed approaches in terms of effectiveness, generation quality, behavioral shift quality, and efficiency.

**Weaknesses:**

The evidence for the effectiveness of the kernel based approach is lacking. According to Figure 3, COLD-kernel approach doesn’t seem to be very effective on several tasks. Section 4.4 (maintaining pluralistic views) seems to be an afterthought to hide this weakness, but it seems a different task than shifting behavior, which is the main claimed goal of the paper.

**Questions:**

How important are the number of examples in the quality of approximations? It seems that for certain tasks, the number of examples does not influence the results while for others the difference in accuracy is significant and sometimes more examples even hurts performance. What are your intuitions?

Does “Base” in Table 3, 4, and 6 refer to no in context examples and no training? That seems to be the weakest baseline across all the methods you have considered as baseline. Why not compare it to DiffMean or ReFT?

Can you provide more intuition about why the kernel based steering preserves subgroup distributional properties better than the finite difference method?

---

> ### Author Response · Authors · 2025-11-21
> **Author Response to Reviewer xSG1 (1/3)**
>
> We are delighted by the reviewer’s positive support of our work. They recognized the significance of our contributions, technical soundness, and thorough experimental evaluation. They also raise very valuable concerns that help us think deeper and improve various aspects of our exposition so as to clarify and address these concerns. We provide our responses to those concerns below and will also revise our paper to incorporate our discussions.
>
> > The evidence for the effectiveness of the kernel based approach is lacking. According to Figure 3, COLD-kernel approach doesn’t seem to be very effective on several tasks.
>
> While COLD-Kernel does underperform relative to COLD-FD, it is still often effective beyond the OpinionsQA setting. In the CAA selection results (Table 2), COLD-Kernel achieves the best performance for steering Llama-2-7b-hf toward myopic-reward behavior, and for steering Llama-2-7b-chat-hf toward hallucination and sycophancy in the positive-only regime. In fact, across all CAA behaviors in the positive-only setting, COLD-Kernel attains the strongest overall average rank (1.47 vs. 2.00 for COLD-FD). Furthermore, in the generation setting (Table 3), COLD-Kernel achieves the best performance on two out of six behaviors. Taken together, these results show that despite its simplicity and efficiency, COLD-Kernel regularly provides meaningful steering gains, and in several cases matches or surpasses more complex alternatives.
>
> > Section 4.4 (maintaining pluralistic views) seems to be an afterthought to hide this weakness, but it seems a different task than shifting behavior, which is the main claimed goal of the paper.
>
> We politely disagree with the characterization that Section 4.4 is an afterthought. Pluralistic steering is a central and motivated application of behavioral steering—indeed, prior work on pluralistic alignment [1] identifies the ability to steer models toward multiple valid viewpoints as one of the four core pathways for achieving pluralistic alignment. Our intention in Section 4.4 is to demonstrate precisely this broader capability of COLD. The OpinionsQA benchmark provides multiple-choice distributions representing opinion patterns across demographic groups. By optimizing a cross-entropy loss over these distributions, we show that COLD can steer models toward the preferences of different populations. While Table 2 focuses on binary-choice behavioral steering, Section 4.4 extends the same framework to distributional steering over diverse demographic viewpoints. Thus, rather than being tangential, Section 4.4 highlights the flexibility and generality of COLD: it supports steering both discrete behaviors and pluralistic, group-conditioned opinion distributions at inference time, which is a key strength of the proposed approach.
>
> > How important are the number of examples in the quality of approximations? It seems that for certain tasks, the number of examples does not influence the results while for others the difference in accuracy is significant and sometimes more examples even hurts performance. What are your intuitions?
>
>
> Empirically, we find that the number of examples can matter, but its impact is moderate and task-dependent. As shown in Figures 1, 3, and 4, COLD-FD and COLD-Kernel often exhibit relatively stable performance as we vary the number of examples, with some tasks showing clear gains and others showing only minor changes. This contrasts with baselines such as DiffMean and ReFT, whose performance curves are typically more sensitive and volatile as the number of examples changes. In addition, when we fix the budget to 50 examples (Table 1), our methods outperform these baselines in almost all cases, indicating that our approximations are more effective even at the same data scale.
>
> Regarding more examples can sometimes slightly hurt performance on certain tasks, our intuition is that this is not due to a noisier gradient estimate per se, but rather due to the limitations of using a single shared one-step update direction. In COLD, the steering signal is derived from the average gradient over all examples (or its finite-difference counterpart). As we add more examples, especially if they are heterogeneous or partially conflicting, the average gradient direction can shift in a way that is not perfectly aligned with the evaluation metric for a fixed single-step update (with fixed step size η). In such cases, the “global” one-step update may implicitly trade off performance across different sub-populations of examples, and this can occasionally lead to small decreases in accuracy as N increases.
>
> *(continued in 2/3)*

---

> ### Author Response · Authors · 2025-11-21
> **Author Response to Reviewer xSG1 (2/3)**
>
> *(continued from 1/3)*
>
> Overall, we view this as a natural consequence of the expressive limits of a single, global, one-step approximation, rather than an inherent instability of the method. Importantly, this behavior is not the norm in our experiments, and in most settings, performance is flat or improves with more examples. We agree that this points to an interesting direction for future work: designing efficient methods that approximate multi-step optimization trajectories or incorporate example reweighting/clustering. Such extensions could better capture the underlying behavioral signal while further reducing sensitivity to the exact number and composition of examples.
>
>
> > Does “Base” in Table 3, 4, and 6 refer to no in context examples and no training? That seems to be the weakest baseline across all the methods you have considered as baseline. Why not compare it to DiffMean or ReFT?
>
>
> Yes, ‘base’ here means the model with no in-context examples and steering, as in Table 2. Since we had already established that COLD can outperform multiple baselines (such as ReFT (vector) and DiffMean) in steering desirable behavior for the selection and generation tasks in Tables 2 and 5 respectively, we decided to omit them in the other tables for brevity to highlight the applicability of our approach in different settings. However, for completeness, we have now included the baselines of DiffMean and ReFT(vector) below for Tables 3, 4, and 6. Note that DiffMean cannot be applied to OpinionsQA since it does not contain a clear pairwise set of positive-negative pairs of behavior. We find that COLD consistently outperforms the baselines except in the case of OpinionsQA, where we find that ReFT performs slightly better than COLD-Kernel. The low performance of COLD-FD at $\epsilon$ = 1e-6 suggests that, in this setting, the activations are particularly sensitive to low-order parameter changes, which get captured by the learned parameters in ReFT. Nevertheless, the simple and efficient COLD-Kernel still closely matches this performance of bulky parameter-tuning, highlighting the strengths of our approach.
>
>
> **Table 3 (hallucination selection for Mistral and Gemma):**
> | | pair | pos |
> |:--|:--|:--|
> | Gemma-2-9B | | |
> | Base | 0.64 | 0.64 |
> | DiffMean | 0.64 | - |
> | ReFT(vector) | 0.64 | 0.64 |
> | COLD-FD | 0.70 | 0.74 |
> | Mistral-7B | | |
> | Base | 0.62 | 0.62 |
> | DiffMean | 0.80 | - |
> | ReFT(vector) | 0.80 | 0.80 |
> | COLD-FD |  0.88 | 0.78 |
>
>
> **Table 4 (generation in CAA):**
> | |   coais |   corr |   hallu |   mr |   ref |   surv |   syco |
> |:---|--:|--:|--:|--:|--:|--:|--:|
> | Llama-2-7b-hf | | | | | | |
> | Base | 4.30 | 3.80 | 5.98 | 4.84 | 3.16 | **4.84** | **4.32** |
> | DiffMean | **5.33** | 3.08 | 7.2 | 5.02 | 3.64 | 4.76 | 4.15 |
> | ReFT(vector) | 3.92 | 2.36 | 7   | 5.38 | 3.88 | 4.66 | 4.24 |
> | COLD-FD | 3.94 | 2.58 | **7.22** | **5.18** | **4.50** | 4.36 | 4.06 |
> | COLD-Kernel | 4.36 | **3.84** | 6.04 | 4.53 | 2.80 | 4.76 | 3.68 |
> | Llama-2-7b-chat-hf | | | | | | |
> | Base | 0.28 | 3.82 | 2.98 | 1.98 | 4.88 | 5.26 | 0.92 |
> | DiffMean |0.3 |  4.4 |    2.64 |    2.08 |      5.5 |        6.04 |    0.81 |
> | ReFT(vector) |  0.14 | 4.46 |    2.92 |   **2.66** |   **5.2** |        **6.22** |    0.69 |
> | COLD-FD | **0.82** | **5.06** | **3.32** | 2.62 | 4.92 | 6.20 | **1.23** |
> | COLD-Kernel | 0.20 | 3.86 | 3.30 | 2.22 | **5.20** | 5.40 | 0.96 |
>
>
> **Table 6 (generation in OpinionsQA):**
>
> | | | Political party | | Race | | | | Sex | |
> |:--|:--|:--|:--|:--|:--|:--|:--|:--|:--|
> |  | |   Democrat |   Republican |   Asian |   Black |   Hispanic |   White |  Female | Male |
> | Base | KL   |      1.17 | 1.09 |    0.91 |    1.12 |  0.9  |    1.07 |    1.04 |  1.08 |
> | Base | TV  |      0.49 | 0.46 |    0.45 |    0.48 |  0.44 |    0.46 |    0.45 |  0.46 |
> | ReFT(vector) | KL       |      0.64 | 0.58 |    0.45 |    0.56 |  0.44 |    0.55 |    0.64 |  0.55 |
> | ReFT(vector) | TV      |      0.42 | 0.39 |    0.36 |    0.39 |  0.33 |    0.38 |    0.39 |  0.37 |
> | COLD-FD | KL    |      2.12 | 1.67 |    1.45 |    1.92 |  1.77 |    1.74 |    1.9  |  1.74 |
> | COLD-FD | TV   |      0.67 | 0.63 |    0.53 |    0.67 |  0.61 |    0.62 |    0.64 |  0.62 |
> | COLD-Kernel | KL  |      0.72 | 0.69 |    0.65 |    0.57 |  0.47 |    0.64 |    0.71 |  0.66 |
> | COLD-Kernel | TV |      0.47 | 0.44 |    0.44 |    0.42 |  0.36 |    0.42 |    0.44 |  0.43 |

---

> > ### Author Response · Authors · 2025-11-21
> > **Official Response to Reviewer xSG1 (3/3)**
> >
> > > Can you provide more intuition about why the kernel based steering preserves subgroup distributional properties better than the finite difference method?
> >
> >
> > We also find this result to be quite intriguing. While the effectiveness of kernel approximation can be dataset-specific (for example, in myopic-reward behavior in CAA or OpinionsQA), the reviewer’s comment has pushed us to explore the cases where a unit kernel can be very effective.
> >
> >
> > First, we investigate why a unit kernel can be effective in general. Specifically, we observe that the approximation $\langle \nabla_{\theta} Z(x_i), \nabla_{\theta} Z(x_j) \rangle \approx 1$ can hold when the parameter gradients of a subnetwork are approximately the same (up to scaling) across inputs in a dataset. This occurs when the per-example gradient vectors are highly aligned or dominated by a single common direction. In such cases, each entry is roughly equal to the product of two similar norms, and after normalization, the resulting kernel closely resembles a unit (all-ones) kernel. Since all the inputs in the dataset are designed to elicit the same underlying behavior, we can expect the gradients with respect to the model’s parameters to be highly aligned. This is based on the assumption that the model internally encodes the relevant high-level concept (such as specific behaviors) in a consistent and linear way across inputs, which is often called the linear representation hypothesis [2,3]. In other words, the directions in activation or gradient space corresponding to a particular concept are similar across different inputs that express the concept. This explains why the kernel, computed as the inner product of per-example gradients, can be well-approximated by a unit (all-ones) matrix: each input contributes a gradient pointing along the same underlying conceptual direction, making them appear nearly identical in the kernel space after normalization.
> >
> >
> > Secondly, we note that a finite difference approximation can be less effective when the loss function is more sensitive to changes below the epsilon value (=1e-6) considered in the finite difference approach. We choose a fixed epsilon for all experiments to show the generalizability of our approach but task-specific values may give higher performance. Since subgroup distributional properties involve a partial cross entropy over multiple choices, it can be more sensitive to smaller changes in the input than considered by the finite difference approach, while the behavior is dominated by a single vector, which is exploited by the unit kernel approach.
> >
> >
> > [1] Sorensen, Taylor, et al. "A roadmap to pluralistic alignment." ICML 2024.
> >
> >
> > [2] Park, Kiho, Yo Joong Choe, and Victor Veitch. "The linear representation hypothesis and the geometry of large language models." ICML 2024.
> >
> >
> > [3] Arditi, Andy, et al. "Refusal in language models is mediated by a single direction." NeurIPS 2024.

---

### Author Response · Authors · 2025-11-26
**Summary of revisions based on the reviews**

We would like to thank all the reviewers for their time and effort in reviewing our manuscript. They have highlighted the technical soundness/motivation of our approach (xSG1, toPp, QoRx, wgtt) and the comprehensiveness of the experimental evaluation (xSG1, toPp, QoRx, wgtt). However, they also raised some concerns that we have thoroughly addressed in the rebuttal and have updated the draft to reflect this feedback in purple color. In particular,
- **Writing clarifications:** We have clarified our claim in lines 40-41 to point towards ReFT and not DiffMean (toPp) and improved the motivation of the pluralistic steering experiments in lines 408-409 (xSG1). To ease the findings of Table 2, we have included an average rank of each method over different behaviors (toPp). In addition, we have also clarified our statement on COLD-Steer’s space complexity in lines 226-227 (wgtt) and now provide a preliminary experiment in Table 9 to motivate future implementations to reduce the complexity.
- **Extended discussion:** We have extended our related work to include Arditi et al., 2024; Rodriguez et al., 2024; and Vu & Nguyen, 2025 (toPp). In particular, we note that these papers are either complementary or contemporaneous to ours. We have also provided a detailed discussion to motivate and explain the effectiveness of the unit kernel approximation in Appendix B.1 (toPp, xSG1, wgtt, QoRx), where we show that unit approximation has a deeper connection with the linear representation hypothesis. In addition, we also provide a discussion on failure cases of COLD-FD in Appendix B.2 (xSG1).
- **Missing baselines (xSG1, toPp):** We add baseline results in Table 3 and include full baseline tables for Tables 4 and 6 as Tables 10 and 11. We will replace Table 4 with 10 and 6 with 11 upon having the extra page. These results further strengthen our claims of the effectiveness of our approach in different settings.
- **Sensitivity experiments:** We include the method’s sensitivity with respect to the steering strength (toPp, wgtt, QoRx) and target steering layers (wgtt, QoRx) in Tables 14 and 15. This helps us to further motivate the hyperparameter choices in our work.
- **Other LLMs (toPp, QoRx):** We extend our analysis on a new LLM: Qwen-2.5-7B-Instruct and provide its results on the selection and generation task in Tables 16 and 17. In addition, we provide the generation result for Mistral-7B-Instruct-v0.1 model in Table 18. This generalizes our findings across models.
- **Effect of steering on generated tokens (toPp):** In Table 19, we analyze the effect of steering on every generated token instead of just the prompt that we did earlier. Results show inconsistent gains and thus motivate our choice of only steering on the prompt for efficiency reasons.

Our rebuttal was acknowledged by the reviewer wgtt and they found it to effectively address their concerns, concluding that the paper is above the acceptance threshold. We thus commit to incorporating these changes in the final version of the paper and including them in the main paper with the extra page available to us upon acceptance.

---

### Meta-Review · Area_Chair_7W9m · 2026-01-06

**Summary:**

The paper makes a solid contribution by unifying contrastive and parameter-tuning steering methods under a gradient-based framework, demonstrating strong sample efficiency. The rebuttal effectively addressed most reviewer concerns with additional experiments, baselines, and theoretical justification. The remaining limitations (large-scale evaluation, untargeted behavior analysis) are reasonable scope boundaries rather than fundamental flaws.

**Reviewer Concerns:**

The authors provided a compelling theoretical justification for the unit kernel approximation by connecting it to the linear representation hypothesis, explaining that gradient vectors are naturally aligned when examples encode the same behavior. Missing baselines (DiffMean, ReFT) were added for Tables 3, 4, and 6, and experiments were extended to a fourth model family (Qwen-2.5-7B-Instruct). Comprehensive sensitivity analyses for steering layer and n multiplier were provided, validating the hyperparameter choices. The authors also justified their prompt-only steering strategy with empirical comparisons showing it performs comparably or better than steering all generated tokens.

While the authors argue COLD is model-agnostic with predictable scaling, no empirical evidence on models >9B parameters was provided. This remains a limitation but is acceptable given computational constraints.

**Reviewer Scores:**

Reviewer **xSG1**: Would likely maintain 8 - concerns were well-addressed, and they were already supportive.

Reviewer **toPp**:  Would likely increase to 5-6 - most technical concerns were thoroughly addressed with additional experiments and baselines, though some theoretical depth remains limited.

Reviewer **wgtt**: Explicitly acknowledged concerns were addressed and confirmed score of 6, stating the paper is "above the acceptance threshold."

Reviewer **QoRx**: Would likely maintain 6 - key concerns about sensitivity and theoretical justification were addressed, though scalability evidence remains limited.

---

### Decision · Program_Chairs · 2026-01-26

Accept (Poster)